# Generalization Bounds for Estimating Causal Effects of Continuous Treatments

Xin Wang      Shengfei Lyu      Xingyu Wu      Tianhao Wu      Huanhuan Chen[*]

University of Science and Technology of China
{wz520, saintfe, xingyuwu, wutianhao8888}@mail.ustc.edu.cn
hchen@ustc.edu.cn

## Abstract

We focus on estimating causal effects of continuous treatments (e.g., dosage in medicine), also known as dose-response function. Existing methods in causal inference for continuous treatments using neural networks are effective and to some extent reduce selection bias, which is introduced by non-randomized treatments among individuals and might lead to covariate imbalance and thus unreliable inference. To theoretically support the alleviation of selection bias in the setting of continuous treatments, we exploit the re-weighting schema and the Integral Probability Metric (IPM) distance to derive an upper bound on the counterfactual loss of estimating the average dose-response function (ADRF), and herein the IPM distance builds a bridge from a source (factual) domain to an infinite number of target (counterfactual) domains. We provide a discretized approximation of the IPM distance with a theoretical guarantee in the practical implementation. Based on the theoretical analyses, we also propose a novel algorithm, called **A**verage **D**ose-response esti**M**at**I**on via re-weigh**T**ing schema (ADMIT). ADMIT simultaneously learns a re-weighting network, which aims to alleviate the selection bias, and an inference network, which makes factual and counterfactual estimations. In addition, the effectiveness of ADMIT is empirically demonstrated in both synthetic and semi-synthetic experiments by outperforming the existing benchmarks.

## 1 Introduction

Causal inference, which estimates causal effects from observational data, is a fundamental problem in identifying causal reactions between actions/treatments and effects. It facilitates the decision-making process in a wide variety of domains, such as economics [1], public policy [2], medicine [3, 4] and advertising [5, 6]. Specifically, causal inference aims to estimate the causal effects of taking different treatments (e.g., whether to take a medication or not), where neural network-based methods have made a breakthrough in the binary setting [7, 8, 9, 10]. In real-world applications, e.g. a medical experiment, treatments usually cannot be simplified as binary treatments since the dosage is a critical factor for the effect of the medication. Compared with causal inference for binary treatments, causal inference for continuous treatments, also known as dose-response estimation, deals with an infinite number of treatments, resulting in an infinite number of counterfactuals, far more than one counterfactual in the binary setting.

The success of neural networks on causal inference for binary treatments [11, 12, 13] can be attributed in part to the alleviation of selection bias, which leads to unreliable counterfactual inference since the covariates of subpopulations receiving different treatments are unbalanced due to non-randomized treatments among them. Specifically, the two distributions of the treated and control groups can

---

[*]Corresponding author

36th Conference on Neural Information Processing Systems (NeurIPS 2022).

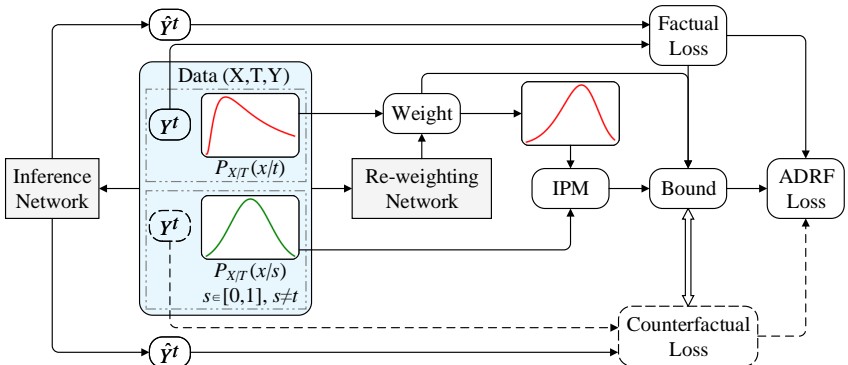

Figure 1: The framework of estimating ADRF using a re-weighting schema and IPM. The counterfactual loss cannot be computed since counterfactual outcomes are not unobserved. In the figure, unobservable variables and incomputable results are indicated by dotted rounded boxes and infeasible computations are indicated by dashed arrows. We learn sample weights using a re-weighting network to make the distributions of the subpopulation receiving treatment $t$ and another subpopulation receiving treatment $s \in [0, 1], s \neq t$ be closer in the IPM distance, i.e., more similar, which results in alleviating selection bias in the continuous treatment setting. We also learn an inference network to make factual and counterfactual estimations. Then we exploit re-weighted factual loss and the IPM distance to bound the counterfactual loss. Finally, we combine the factual loss and the bound of the counterfactual loss as the ADRF loss, which is minimized to simultaneously learn the re-weighting network and the inference network.

be balanced using Integral Probability Metric (IPM) [11] or re-weighting schemes [12, 13]. In the continuous setting, since the control groups are potentially infinite, these standard methods for alleviating selection bias in binary treatments cannot be easily extended to mitigate selection bias among a theoretically infinite number of subpopulations.

Currently, Schwab et al. [14] propose the Dose Response Network (DRNet) that extends "two heads", which are two outcomes in a neural network for making factual and counterfactual estimations in the binary setting, to "multiple heads" for a continuous dosage by discretizing the dosage into multiple intervals. Several regularization schemes in DRNet, such as distribution matching [7], propensity dropout [15] and matching on balancing scores [16, 17], are exploited to handle selection bias. However, these schemes are heuristic and do not provide theoretical evidence to back up their intuition.

To theoretically support the alleviation of selection bias, this paper derives an upper bound on the counterfactual loss during estimation of the average dose-response function (ADRF). The key points of the derivation lie in the re-weighted factual loss obtained from observational data and an IPM term that measures the distance between the factual (re-weighted) and counterfactual distributions, as illustrated in Figure 1. The infinite number of counterfactual distributions is the main challenge since the distance cannot be calculated with finite samples in practice. In the practical implementation, we provide a discretized approximation of the IPM term and prove that the difference between its approximate and true values converges to zero in probability under the assumption that the distributions of subpopulations receiving different treatments shift smoothly. On the basis of the derived upper bound and its implementation with a theoretical guarantee, we propose a novel algorithm called **A**verage **D**ose-response esti**M**at**I**on via re-weigh**T**ing schema (ADMIT). ADMIT jointly learns an inference network and a re-weighting neural network to reduce selection bias while making factual and counterfactual estimations. The effectiveness of ADMIT is empirically confirmed by outperforming the existing benchmarks on both synthetic and semi-synthetic experiments in the continuous treatment setting.

Our contributions in this paper are 4-fold: (1) to the best of our knowledge, this is the first study that provides a generalization bound for estimating ADRF, alleviating selection bias; (2) we provide a discretized approximation of the IPM distance with a theoretical guarantee; (3) we propose an algorithm, ADMIT, capable of reducing selection bias while making factual and counterfactual estimations; (4) we conduct both synthetic and semi-synthetic experiments in the continuous treatment

setting, empirically demonstrating the effectiveness of ADMIT with superiority over the existing benchmarks.

## 2 Related Work

Causal inference for continuous treatments is broadly studied through statistical methods [18, 19, 20, 21]. These methods are often based upon the generalized propensity score (GPS) [22], or the entropy balancing based method [23] that learns the unit weights by incorporating information about known sample moments to achieve covariate balance, e.g., entropy balancing for continuous treatments (EBCT) [24]. Adaptation of neural networks for causal inference in the continuous treatments setting has attracted the attention of research community recently [14, 25, 26]. Schwab et al. [14] present the DRNet to estimate counterfactuals for continuous treatments, which is an extension of the network architecture proposed in [11]. Specifically, DRNet divides a continuous treatment into several successive intervals, and trains one separate head for each interval. Bica et al. [26] point out that DRNet might be inflexible since it cannot determine the divided intervals dynamically and propose SCIGAN that generates counterfactual outcomes for continuous treatments based on a generative adversarial network (GAN) framework. Meanwhile, for the issue that DRNet produces discontinuous ADRF due to interval discretization of continuous treatments, Varying Coefficient Network (VCNet) [25] lets the weights of the prediction head be continuous functions of the treatment to preserve the continuity of ADRF. Moreover, Nie et al. [25] obtain a doubly robust estimator of ADRF by generalizing the targeted regularization proposed in [10], and provide theoretical guarantees for its asymptotic correctness. Unlike VCNet, our work is derived from the generalization error of reducing the selection bias when estimating ADRF. To balance the covariates among infinite subpopulations, we learn re-sampling weights that reduce the IPM distance between observed and counterfactual groups. As such, we derive an upper bound on the estimated counterfactual error and demonstrate experimentally the proposed algorithm ADMIT based on the derived upper bound outperforms GPS, EBCT, DRNet, SCIGAN and VCNet.

See Appendix A for a discussion on research related to causal inference for binary or continuous treatments. Appendix A also introduces the relationships between causal inference for continuous treatments and domain adaptation [27, 28, 29].

## 3 Problem Formulation

Let $\mathcal{X}$ denote the $d$-dimensional space of covariates and $\mathcal{Y}$ the outcome space. We refer to $T$ as a continuous treatment random variable in an interval $\mathcal{T} = [a, b]$. Without loss of generality, we set $\mathcal{T} = [0, 1]$ in this study. Suppose we have an i.i.d. sample of units, indexed by $i = 1, 2, \cdots, n$. Following Rubin's potential outcome framework [30], we postulate the existence of potential outcomes, $Y^t$, for $t \in \mathcal{T}$. For each unit $i$ we observe the treatment $t_i$, and the covariate $\boldsymbol{x}_i$. The observed outcome (denoted as $Y_i$) is defined to be the potential outcome (denoted as $Y_i^{t_i}$) corresponding to the received treatment $t_i$, i.e., $Y_i = Y_i^{t_i}$.

Given an input covariate $\boldsymbol{x}$ and a treatment $t$, we refer to $\mu(\cdot, \cdot)$ as the individual dose-response function, which is defined as:
$$\mu(t, \boldsymbol{x}) := \mathbb{E}[Y^t | \boldsymbol{X} = \boldsymbol{x}]. \tag{1}$$
Our goal is to derive an estimator of the average dose-response function (ADRF):
$$\mu(t) := \mathbb{E}[Y^t]. \tag{2}$$
By the law of iterated expectation, $\mu(t) = \mathbb{E}[\mu(t, \boldsymbol{X})]$. In this study, the estimator of ADRF $\mu(t)$ is based on estimating the individual dose-response function $\mu(t, \boldsymbol{x})$.

The following assumptions have been made to ensure that ADRF is identifiable from observational data.

**Assumption 1.** *(Unconfoundedness) Conditional on the covariates, $\boldsymbol{X}$, the potential outcomes, $Y^t$, are independent of the treatment assignment, $T$,*
$$\{Y^t | t \in [0, 1]\} \perp T | \boldsymbol{X}.$$

This assumption is also known as *no unmeasured confounding* and ensures that the individual dose-response function is identifiable from observational data, i.e., $\mu(t, \boldsymbol{x}) = \mathbb{E}[Y^t | \boldsymbol{X} = \boldsymbol{x}] = \mathbb{E}[Y | \boldsymbol{X} = \boldsymbol{x}, T = t]$.

**Assumption 2.** *(Overlap) There exists some constant $c > 0$, $\forall \boldsymbol{x} \in \mathcal{X}$ and $\forall t \in \mathcal{T}$,*

$$P_{T|\boldsymbol{X}}(t|\boldsymbol{x}) \geq c.$$

This assumption is the condition that every point in the space of covariates has a non-zero conditional probability density of receiving any treatment.

## 4 Theoretical Analyses and Algorithm

This section firstly presents a theorem that the error of estimating ADRF is bounded by the marginal loss comprising the factual and counterfactual losses. The counterfactual loss cannot be calculated from observational data since counterfactual outcomes as the ground truth are unobserved. In the meanwhile, the counterfactual inference might be unreliable due to selection bias. We then derive theoretically optimal weights in alleviating selection bias from the perspective of importance sampling. Although the optimal weights are theoretically guaranteed, they are impractical to learn due to their instability. Under a similar re-weighing schema, a theoretical guarantee is provided to guide the learning of adaptive weights that mitigate the impact of selection bias. We refer the reader to Appendix B for detailed proofs.

### 4.1 Error of estimating ADRF

Let $\hat{\mu}(t)$ be an estimator for $\mu(t)$, and we are interested in estimators with a small expected mean squared error (EMSE) in estimating ADRF,

$$\text{EMSE}(\mu, \hat{\mu}) = \mathbb{E}[(\hat{\mu}(\text{T}) - \mu(\text{T}))^2]. \tag{3}$$

Since $\mu(t)$ cannot be observed from observational data, this study uses hypotheses $f_t \in \mathcal{H}$ to estimate the individual dose-response function $\mu(t, \boldsymbol{x})$ first. For ease of expression, we write $f(t, \boldsymbol{x})$ as $f_t(\boldsymbol{x})$. After that, we combine these hypotheses to estimate $\mu(t)$,

$$\hat{\mu}(t) = \mathbb{E}[f_t(\boldsymbol{X})]. \tag{4}$$

We bound the error $\text{EMSE}(\mu, \hat{\mu})$ by the loss of hypotheses $f_t$, for $t \in \mathcal{T}$, with respect to the corresponding potential outcomes. Let $L : \mathcal{Y} \times \mathcal{Y} \to \mathbb{R}_+$ be a loss function. The expected loss for the treatment and covariate pair $(t, \boldsymbol{x})$ is defined as:

$$l_{f_t}(\boldsymbol{x}) := \mathbb{E}_{Y^t|\boldsymbol{X}}[L(Y^t, f_t(\boldsymbol{X}))|\boldsymbol{X} = \boldsymbol{x}]. \tag{5}$$

**Theorem 1.** *Let $L$ be the squared loss function, i.e., $L(y, y') = (y - y')^2$. For hypotheses $f_t$ of individual dose-response function $\mu(t, \cdot)$ with marginal loss $\epsilon(f_t) = \mathbb{E}[l_{f_t}(\boldsymbol{X})]$, there exists a constant $\sigma_{min} \geq 0$, such that,*

$$\text{EMSE}(\mu, \hat{\mu}) \leq \mathbb{E}_T[\epsilon(f_t)] - \sigma_{min}. \tag{6}$$

Though $\mu(t)$ cannot be observed from observational data, Theorem 1 implies that its expected error can be bounded by the marginal loss $\epsilon(f_t)$, which are related to the prediction loss concerning potential outcomes, i.e., $l_{f_t}(\boldsymbol{x})$. However, the model cannot obtain marginal loss due to the inability to observe the counterfactuals [31]. A re-weighting schema to address this issue is presented next.

### 4.2 Re-weighting via importance sampling

The fundamental problem of estimating ADRF is that it is impossible to observe all potential outcomes for any unit in data. The potential outcome $Y^t$ could only be observed in the subpopulation that received treatment $t$, i.e., $T = t$. We refer to $p_{\boldsymbol{X}|T}(\boldsymbol{x}|t)$ as the distribution of $T = t$ subpopulation, and $p_{\boldsymbol{X}}(\boldsymbol{x})$ as the distribution of population. The two distributions, $p_{\boldsymbol{X}|T}(\boldsymbol{x}|t)$ and $p_{\boldsymbol{X}}(\boldsymbol{x})$, are not equal since $\boldsymbol{X}$ and $T$ are not independent. Their difference leads to the *margin loss* $\epsilon(f_t)$ not being equal to the *conditional loss* of hypothesis $f_t$, $\epsilon(f_t|T = s)$, which is defined as:

$$\epsilon(f_t|T = s) = E_{\boldsymbol{X}|T}[l_{f_t}(\boldsymbol{X})|T = s]. \tag{7}$$

The *conditional loss* $\epsilon(f_t|T = s)$ is the expected loss with respect to the distribution of the subpopulation that received treatment $s$, i.e., $T = s$, and the *marginal loss* $\epsilon(f_t)$ is the expected loss with respect to the distribution of the population. When the treatments $t$ and $s$ are equal, the *conditional loss* $\epsilon(f_t|T = s)$ is called *factual loss*; otherwise, it is called *counterfactual loss*. The *counterfactual loss* is infeasible since the potential outcome $Y^t$ could only be observed in the subpopulation who receive the treatment $t$. A minimizer $f_t$ of $\epsilon(f_t)$ is required to minimize the error of estimating ADRF according to Theorem 1, which may be different from a minimizer of $\epsilon(f_t|T = t)$ since $p_{\boldsymbol{X}|T}(\boldsymbol{x}|t) \neq p_{\boldsymbol{X}}(\boldsymbol{x})$. Importance sampling is introduced to estimate $\epsilon(f_t)$ in this work, which aims to estimate $\mathbb{E}_{\boldsymbol{x} \sim g(\boldsymbol{x})}[h(\boldsymbol{x})]$ of a particular distribution $g(\boldsymbol{x})$ while only samples generated from a different distribution $q(\boldsymbol{x})$ are available. The basic idea is to rewrite the expectation as:

$$\mathbb{E}_{\boldsymbol{x} \sim g(\boldsymbol{x})}[h(\boldsymbol{x})] = \mathbb{E}_{\boldsymbol{x} \sim q(\boldsymbol{x})}\left[h(\boldsymbol{x}) \cdot \frac{g(\boldsymbol{x})}{q(\boldsymbol{x})}\right], \tag{8}$$

where $\frac{g(\boldsymbol{x})}{q(\boldsymbol{x})}$ is called sampling weight. To estimate $\epsilon(f_t)$ based on the *factual loss* $\epsilon(f_t|T = t)$, a weight $w(x)$ is utilized for each unit to define a *re-weighted factual loss*:

$$\epsilon_w(f_t|T = t) = \mathbb{E}_{\boldsymbol{X}|T}[w(\boldsymbol{X})l_{f_t}(\boldsymbol{X})|T = t]. \tag{9}$$

By Equation (8), the *re-weighted factual loss* is equal to the *marginal loss* when $w(\boldsymbol{x}) = w^*(\boldsymbol{x})$, where

$$w^*(\boldsymbol{x}) = \frac{p_{\boldsymbol{X}}(\boldsymbol{x})}{p_{\boldsymbol{X}|T}(\boldsymbol{x}|t)} = \frac{p_T(t)}{p_{T|\boldsymbol{X}}(t|\boldsymbol{x})}. \tag{10}$$

The term $p_{T|\boldsymbol{X}}(t|\boldsymbol{x})$ is the conditional density of receiving a treatment $t$ given the covariate $\boldsymbol{x}$, which is also known as the *Generalized Propensity Score* (GPS) [22, 32]. In binary treatment cases, the propensity score could balance the distributions of covariates between the treated and control groups [33]. Hirano and Imbens [32] stress that the GPS has a similar balancing property in continuous treatment cases. Fong et al. [20] propose the covariate balancing generalized propensity score methodology by defining a stabilized weight based on the GPS. Interestingly, the weight defined in [20] is exactly the same as $w^*(\boldsymbol{x})$ derived from the sample weight via importance sampling.

The weight $w^*(\boldsymbol{x})$ based on the GPS is theoretically optimal in expectation [22] but may be impractical to obtain. On the one hand, the approaches based on the propensity score may be unstable in the estimations [34, 35]. What's worse, this issue may be amplified in continuous treatment cases since the conditional density is difficult to estimate accurately [31]. On the other hand, it could be proved that a large sampling weight $\frac{g(\boldsymbol{x})}{q(\boldsymbol{x})}$ may result in a high variance in the estimations, which could lead to a poor finite sample performance. Therefore, the following subsection focuses on learning adaptive weights to balance observed covariates across subpopulations that received different treatments and giving a generalization bound of the *marginal loss*.

### 4.3 Generalization bound of the *marginal loss*

The imbalance of covariates across subpopulations that received different treatments caused by selection bias is closely connected with covariate shift in domain adaptation. This study alleviates this imbalance using an IPM distance, which is similar to the distribution distance metrics in domain adaptation, and gives a generalization bound of the *marginal loss* when using adaptive weights in Equation (9).

The IPM is a class of metrics that measure the distance between two probability distributions, which is defined as:

$$\text{IPM}_{\mathcal{G}}(p, q) = sup_{g \in \mathcal{G}} \left| \int g(\boldsymbol{x})(p(\boldsymbol{x}) - q(\boldsymbol{x}))d\boldsymbol{x} \right|, \tag{11}$$

where $\mathcal{G}$ is a function family of functions $g : \mathcal{X} \to \mathcal{R}$. For simplicity, we abbreviate $p_{\boldsymbol{X}|T}(\boldsymbol{x}|t)$ and $p_{\boldsymbol{X}|T}(\boldsymbol{x}|t)w(\boldsymbol{x})$ as $p_t$ and $p_t^w$, respectively. We first state a Lemma bounding the difference between the *re-weighted factual loss*, $\epsilon_w(f_t|T = t)$, and the *counterfactual loss*, $\epsilon(f_t|T = s)$, where $s \neq t$.

**Lemma 1.** *Let $\mathcal{G}$ be a family of functions $l : \mathcal{X} \to \mathcal{R}$. Assume the per-unit expected loss function $L(f, f') \in \mathcal{G}$ for all $f, f' \in \mathcal{H}$. Then for any $s \in [0, 1]$ and $s \neq t$, we have:*

$$\epsilon(f_t|T = s) \leq \epsilon_w(f_t|T = t) + \text{IPM}_{\mathcal{G}}(p_s, p_t^w). \tag{12}$$

Based on Lemma 1, we could give a generalization bound of the *marginal loss* through the *re-weighted factual loss* as follows.

**Theorem 2.** *Let* $\text{IPM}_{max} = max_{s \in [0,1]}\{\text{IPM}_{\mathcal{G}}(p_s, p_t^w)\}$. *The following inequality holds under the conditions of Lemma 1,*

$$\epsilon(f_t) \leq \epsilon_w(f_t | T = t) + \text{IPM}_{max}. \tag{13}$$

Theorem 2 shows that the *marginal loss* is upper bound by the *re-weighted factual loss* and an IPM distance. The first term could be estimated with observed samples. The second term is defined based on the distance between the two probability distributions of the subpopulation receiving treatment $t$ and another subpopulation. Choosing a family of norm-1 reproducing kernel Hilbert space (RKHS) functions leads to IPM being the Maximum Mean Discrepancy (MMD) metric [36]. When involving discrete domains, there may have samples $\{\boldsymbol{x}_1^p, \cdots, \boldsymbol{x}_n^p\} \sim p$ and $\{\boldsymbol{x}_1^q, \cdots, \boldsymbol{x}_m^q\} \sim q$, an estimator of the MMD distance between $p$ and $q$ could be obtained as follows:

$$\widehat{\text{MMD}}_k^2(p, q) := \frac{1}{n^2}\sum_{i=1}^{n}\sum_{j=1}^{n}k(\boldsymbol{x}_{\boldsymbol{i}}^p, \boldsymbol{x}_{\boldsymbol{j}}^p) - \frac{2}{mn}\sum_{i=1}^{n}\sum_{j=1}^{m}k(\boldsymbol{x}_{\boldsymbol{i}}^p, \boldsymbol{x}_{\boldsymbol{j}}^q) + \frac{1}{m^2}\sum_{i=1}^{m}\sum_{j=1}^{m}k(\boldsymbol{x}_{\boldsymbol{i}}^q, \boldsymbol{x}_{\boldsymbol{j}}^q), \tag{14}$$

where $k$ denotes a differentiable kernel, e.g., the Gaussian RBF kernel. A certain number of samples are needed to estimate the MMD as shown in Equation 14. When the treatment is discrete and finite, all the choices of the treatment can be observed in the samples and thus one can estimate the MMD from these samples. However, when the treatment is continuous, the samples that received some treatment $t$ may be unavailable since only a finite number of samples are observed while the choice of $t$ is infinite. In other words, the IPM term in Theorem 2 involves continuously shifting domains (for each $s \in [0, 1]$), which cannot be estimated empirically. To overcome this challenge, we bound the difference between the $\text{IPM}_{max}$ and its discretization under the assumption that the probability distributions of subpopulations that received different treatments shift smoothly.

**Assumption 3.** *Let $p_{t_1}$ and $p_{t_2}$ denote the conditional probability densities of subpopulations that received treatment $t_1$ and $t_2$, respectively. We assume that there is a constant $\alpha$ such that the following inequality holds for all $t_1, t_2 \in [0, 1]$:*

$$\text{IPM}_{\mathcal{G}}(p_{t_1}, p_{t_2}) \leq \alpha |t_1 - t_2|. \tag{15}$$

It could be proved that the minimum $\alpha$ that meets the conditions of Assumption 3 is $max_{t \in [0,1]}\{\lim_{\delta \to 0}\}\frac{IPM(p_t, p_{t+\delta})}{\delta}$ according to the triangle inequality for the IPM. Intuitively, $\alpha$ in Assumption 3 indicates the maximum rate of the probability distribution shift of subpopulations, and it is easy to find a constant $\alpha$ that satisfy the condition since the IPM term is bounded when the outcome of the hypothesis $f_t$ is finite, e.g., survival years (outcome) after taking some medicine (treatment). Therefore, Assumption 3 is practical and feasible in real-world applications. When we set a continuous treatment $t$, it is generally assumed that the conditional probability of $t$ given $x$ smoothly shift [26]. For instance, the difference in the probability of taking a similar dose given an age is small if the dose of a particular medicine depends on the age of the patient. Therefore, the age distributions of groups taking similar doses vary smoothly, or in other words, it holds most of the time that $\alpha$ is relatively small.

Based on Assumption 3, the following Lemma states that the $\text{IPM}_{max}$ corresponding to continuously shifting domain (for each $t \in [0, 1]$) could be approximated as an IPM term based on discrete domains (for each $t \in \{t_1, \cdots, t_n\}$).

**Lemma 2.** *Suppose we have $n$ i.i.d. sample of units, and the $i$th unit received a treatment $t_i \sim p(t)$. This study assumes Assumption 3 holds for a constant $\alpha$. Then we have,*

$$\text{IPM}_{max} \leq max_{i \in \{1, \cdots, n\}}\{\text{IPM}_{\mathcal{G}}(p_{t_i}, p_t^w)\} + O_p\left(\frac{\alpha}{\sqrt[3]{n}}\right). \tag{16}$$

The RHS of Equation 16 bounds the worst case of $\text{IPM}_{max}$ by a discrete set $\{\text{IPM}_{\mathcal{G}}(p_{t_i}, p_t^w) | i \in \{1, \cdots, n\}\}$ since all $\alpha_i$ are enlarged to $\alpha$ during the proof of Lemma 2, where $\alpha_i = max_{t \in [t_i, t_{i+1}]}\frac{\text{IPM}(p_t, p_{t_i})}{t - t_i}$. Therefore, the difference between the $\text{IPM}_{max}$ and its discretization will be generally smaller than Lemma 2 displays. To empirically estimate $\text{IPM}_{\mathcal{G}}(p_{t_i}, p_t^w)$, the following Lemma states that a point $t_i$ could be replaced by its neighbourhood $[t_i, t_i + \delta]$.

**Algorithm 1** ADMIT: Average dose-response estimation via re-weighting schema

---

**Input:** Observed samples $\mathcal{D} = \{(\boldsymbol{x_i}, s_i, Y_i)|i = 1, \cdots, n\}$, loss function $L(\cdot, \cdot)$,function family $\mathcal{G}$ for IPM, batch size $n_b$, neighbourhood size $\delta$. Initialized parameters: $\theta = [\theta_\phi, \theta_h, \theta_\omega]$

1: **while** $\theta$ not converged **do**
2:     Sample mini-batch $\{(\boldsymbol{x}_{i_1}, s_{i_1}, y_{i_1}), \cdots, (\boldsymbol{x}_{i_{n_b}}, s_{i_{n_b}}, y_{i_{n_b}})\}$ from $\mathcal{D}$
3:     Calculate the weight of each unit: $w_{i_j} = \omega(s_{i_j}, \phi(x_{i_j}))$
4:     **for** $l = 1$ to $L = \lceil 1/\delta \rceil$ **do**
5:         **for** $k = 1$ to $L$ and $k \neq l$ **do**
6:             $\text{IPM}_\mathcal{G}(p_{\Delta k}, p_{\Delta l}^w) = \text{IPM}_\mathcal{G}(\{w_{i_j} \cdot \boldsymbol{x}_{i_j}\}_{s_{i_j} \in [(l-1)\delta, l\delta]}, \{\boldsymbol{x}_{i_h}\}_{s_{i_h} \in [(k-1)\delta, k\delta]})$
7:         **end for**
8:         $\text{IPM}_{\Delta max}^l = max_{k \in \{1, \cdots, L\}, k \neq l} \{\text{IPM}_\mathcal{G}(p_{\Delta k}, p_{\Delta l}^w)\}$
9:     **end for**
10:    Calculate the gradients of the IPM term:
      $g_1 = \Delta_{\theta_\omega} \frac{1}{L} \sum_l \text{IPM}_{\Delta max}^l$
      $g_2 = \Delta_{\theta_\phi} \frac{1}{L} \sum_l \text{IPM}_{\Delta max}^l$
11:    Calculate the gradients of the *re-weighted factual loss*:
      $g_3 = \Delta_{\theta_h} \frac{1}{n_b} \sum_j w_{i_j} \cdot L(h(s_{i_j}, \phi(\boldsymbol{x}_{i_j})), y_{i_j})$
      $g_4 = \Delta_{\theta_\phi} \frac{1}{n_b} \sum_j w_{i_j} \cdot L(h(s_{i_j}, \phi(\boldsymbol{x}_{i_j})), y_{i_j})$
12:    Update parameters: $[\theta_\phi, \theta_h, \theta_\omega] \leftarrow [\theta_\phi - \eta(g_2 + g4), \theta_h - \eta g_3, \theta_\omega - \eta g_1]$
13: **end while**

**Output:** Learned representation function $\phi(\cdot)$, outcome function $h(\cdot, \cdot)$, re-weighting function $\omega(\cdot, \cdot)$

---

**Lemma 3.** *Let $p_{\Delta s} = P_{\boldsymbol{X}|T}(\boldsymbol{x}|t \in [s, s + \delta])$ $(0 < \delta < 1)$ denote the conditional density of covariates when $t \in [s, s + \delta]$. Then the following inequality holds under Assumption 3,*

$$\text{IPM}_\mathcal{G}(p_s, p_t^w) \leq \text{IPM}_\mathcal{G}(p_{\Delta s}, p_t^w) + \alpha\delta. \tag{17}$$

Following Lemma 2 and Lemma 3, we bound the *marginal loss* as follows.

**Theorem 3.** *Suppose we have $n$ i.i.d. sample of units, and the $i$th unit received a treatment $t_i$. Let $\text{IPM}_{\Delta max} = max_{i \in \{1, \cdots, n\}} \{\text{IPM}_\mathcal{G}(p_{\Delta t_i}, p_t^w)\}$. This study assumes Assumption 3 holds for a constant $\alpha$. Then, for a neighbourhood size $0 < \delta < 1$ we have,*

$$\epsilon(f_t) \leq \epsilon_w(f_t|T = t) + \text{IPM}_{\Delta max} + O_p\left(\frac{\alpha}{\sqrt[3]{n}}\right) + \alpha\delta. \tag{18}$$

Contrary to the second term in Theorem 2, $\text{IPM}_{\Delta max}$ could be estimated empirically, since we may get samples that received treatment $t \in [t_i, t_i + \delta]$ by choosing an appropriate neighbourhood size $\delta$.

To minimize the error $\epsilon_u$ in estimating the ADRF, Theorem 1 and Theorem 3 suggest that we should minimize the upper bound in Equation (18) with respect to the *re-weighted conditional loss* and the IPM term, which leads us to Algorithm 1 detailed in the following subsection.

### 4.4 Proposed algorithm

Our algorithm, ADMIT, is proposed based on the theoretical results derived in the previous subsections. As illustrated in Figure 2, we use neural networks to build the model, which is mainly composed of three components: a representation network $\phi(\boldsymbol{x})$, a re-weighting network $\omega(t, \phi(\boldsymbol{x}))$, and an inference network $h(t, \phi(\boldsymbol{x}))$.

This work firstly learns a treatment-agnostic representation $z = \phi(\boldsymbol{x})$ of $\boldsymbol{x}$ using all data. The influence of treatment $t$ is different from that of $\boldsymbol{x}$ [11], which might be lost during training if $\phi(\boldsymbol{x})$ is high-dimensional. To address this issue, the varying coefficient model [37, 25] is adopted to build $h(t, \phi(\boldsymbol{x}))$ and $\omega(t, \phi(\boldsymbol{x}))$. Take the former as an example, $h(t, \phi(\boldsymbol{x}))$ is defined as:

$$h(t, \phi(\boldsymbol{x})) = \text{NN}_{\theta_h(t)}(\phi(\boldsymbol{x})), \tag{19}$$

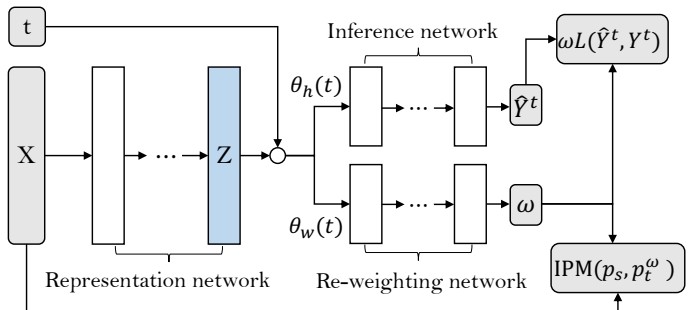

Figure 2: Model architecture of the proposed ADMIT.

where $\mathrm{NN}_{\theta_h(t)}$ is a neural network with parameter $\theta_h(t)$. Thus, the treatment $t$ determines the nonlinear functions in $\mathrm{NN}_{\theta_h(t)}$.

Based on the theoretical analyses aforementioned, ADMIT aims to seek a re-weighting function $\omega$ and an inference function $h$ to minimize the error $\mathrm{EMSE}(\mu, \hat{\mu})$ of estimating the ADRF, using the following objective:

$$\min_\theta \frac{1}{n} \sum_{i=1}^{n} (\omega(t_i, \phi(\boldsymbol{x_i})) \cdot L(h(t_i, \phi(\boldsymbol{x_i})), Y_i) + \max_{j \in \{1, \cdots, n\}, j \neq i} \{\mathrm{IPM}_{\mathcal{G}}(p_{\Delta t_j}, p_{\Delta t_i}^w)\}), \quad (20)$$

where $\theta = [\theta_\phi, \theta_\omega, \theta_h]$. To effectively calculate the IPM, the interval $[0, 1]$ is divided into $\lceil 1/\delta \rceil$ intervals to confirm the neighbourhood of the $i$th unit, and details of the calculation are presented in lines 4 to 9 of Algorithm 1. The model is trained using stochastic gradient descent to minimize the objective. Finally, an estimator of ADRF is built as:

$$\hat{\mu}(t) = \frac{1}{n} \sum_{i=1}^{n} h(t, \phi(\boldsymbol{x_i})). \quad (21)$$

## 5 Experiments

It is difficult to get the ground truth treatment effect on real-world datasets due to the inability to observe the counterfactuals. To deal with this, existing literature [14, 25, 26] often use synthetic/semi-synthetic data for meaningful evaluation. We use one synthetic dataset and two semi-synthetic datasets, News [14, 7] and TCGA [38], to demonstrate the effectiveness of ADMIT. Details of these datasets and implementation can be found in Appendix C.

### 5.1 Experimental setup

**Synthetic data generation.** We generate covariates $\boldsymbol{x} \sim \mathrm{Unif}(\mathbf{0}, \mathbf{1}) \in \mathbb{R}^6$, and the assigned treatments and their corresponding outcomes are generated as follows:

- $t = (1 + \exp(-t'))^{-1}, t' \sim \mathcal{N}(\mu_1, 0.5)$ where

$$\mu_1 = 10 \frac{\sin(\max(x_1, x_2, x_3)) + \max(x_3, x_4, x_5)^3}{1 + (x_1 + x_5)^2} + \sin(0.5x_3)(1 + \exp(x_4 - 0.5x_3))$$
$$+ x_3^2 + 2\sin(x_4) + 2x_5 - 6.5,$$

- $y \sim \mathcal{N}(\mu_2, 0.5)$ where $\mu_2 = \cos(2\pi(t - 0.5))(t^2 + 4\frac{\max(x_1, x_6)^3}{1 + 2x_3^2} \sin(x_4))$.

**Semi-synthetic data generation.** We obtain covariates from two real-world datasets, News [14, 7] and TCGA [38]. To obtain the assigned treatments and their corresponding outcomes, we generate a set of parameters, $\boldsymbol{v}_i = \boldsymbol{u}_i / ||\boldsymbol{u}_i||$ and $i = 1, 2, 3$, where $\boldsymbol{u}_i$ is sampled from a normal distribution $\mathcal{N}(\mathbf{0}, \mathbf{1})$, then

| Method | Simulation | TCGA | News |
|---|---|---|---|
| DRNet | $0.209 \pm 0.0127$ | $0.216 \pm 0.0281$ | $0.274 \pm 0.0194$ |
| SCIGAN | $0.638 \pm 0.0467$ | $0.301 \pm 0.0204$ | $0.707 \pm 0.0074$ |
| VCNet | $0.129 \pm 0.0186$ | $0.139 \pm 0.0107$ | $0.201 \pm 0.0193$ |
| VCNet+EBCT | $0.162 \pm 0.0256$ | $0.126 \pm 0.0234$ | $0.196 \pm 0.0123$ |
| GPS | $0.179 \pm 0.0016$ | $0.196 \pm 0.0011$ | $0.176 \pm 0.0049$ |
| ADMIT | $\mathbf{0.081 \pm 0.0048}$ | $\mathbf{0.071 \pm 0.0074}$ | $\mathbf{0.106 \pm 0.0135}$ |

Table 1: Comparison of ADMIT with (non) neural network-based baselines. Reported performance ($\sqrt{\text{EMSE}}$) of average dose-response estimation on one synthetic and two semi-synthetic datasets. Metrics are reported as Mean $\pm$ Std.

- $t \sim \text{Beta}(\gamma, \beta)$ where $\beta = \frac{\gamma - 1}{d^*} + 2 - \gamma$ and $d^* = \frac{\boldsymbol{v}_3^T \boldsymbol{x}}{2\boldsymbol{v}_2^T \boldsymbol{x}}$,

- $y \sim \mathcal{N}(\mu, 0.5)$ where $\mu = 4(t - 0.5)^2 \times \sin(\frac{\pi}{2} t) \times 2(\max(-2, \exp((\frac{\boldsymbol{v}_2^T \boldsymbol{x}}{\boldsymbol{v}_3^T \boldsymbol{x}} - 0.3))) + 10\boldsymbol{v}_1^T \boldsymbol{x}$.

As discussed in [26], a larger $\gamma$ will result in a higher selection bias. Without specific instruction, we set $\gamma = 2$ in our experiments.

**Baselines and metrics.** For neural network baselines, we compare ADMIT with DRNet [14], VCNet [25] and SCIGAN [26]. With the same settings as in [25], we apply targeted regularization on both DRNet and VCNet, and add a conditional density estimation head for DRNet. For statistical baselines, we compare ADMIT with GPS [32] and entropy balancing for continuous treatments (EBCT) [24]. When evaluating EBCT, we adopt the sample weights learned by EBCT, and take VCNet as its inference network to estimate the ADRF. For metrics, we use Expected Mean Squared Error (EMSE) on the test dataset, where $\text{EMSE} = \frac{1}{N} \sum_{i=1}^{N} (\hat{\mu}(t_i) - \mu(t_i))^2$.

### 5.2 Benchmarks comparison

Table 1 shows that the proposed ADMIT [2] is more accurate in estimating the average dose-response in both synthetic and semi-synthetic datasets. The dimensions of covariates in TCGA and News are around 4000 and 3000, respectively. This result may indicate that ADMIT is well adapted to the high-dimensional data. Moreover, using the sample weights obtained by pre-processing the data with EBCT also enhances the performance on the two semi-synthetic datasets compared with VCNet. This result demonstrates the sample weights learned by EBCT could alleviate the selection bias to some extent. However, the performance boost is smaller than ADMIT, which indicates the superiority of the learned sample weights guided by the derived bound in this study. Additionally, the performance of ADMIT

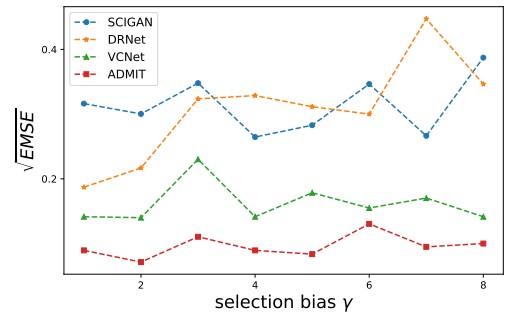

Figure 3: Comparison of four models in terms of their $\sqrt{\text{EMSE}}$ for varying selection bias on TCGA.

may benefit from the design where ADMIT jointly learns the re-weighing network and the inference network in one optimization by minimizing the derived upper bound. This one-step approach may be preferable to two-step approaches, i.e., EBCT, which is demonstrated by Zhang et al. [39].

---

[2]The implementation of ADMIT is available at https://github.com/waxin/ADMIT

### 5.3 Selection bias

As indicated in [26], the variance of the Beta distribution decreases with the increase of $\gamma$ in the semi-synthetic data generation, resulting in higher selection bias. We compare the performance of ADMIT with VCNet, SCIGAN and DRNet on TCGA when varying choices of $\gamma$ from 1 (no bias) to 8 (high bias) to assess the robustness of these methods in different levels of selection bias. We found that ADMIT shows consistent performance and outperforms the three baselines across the entire range of evaluated selection bias. This result demonstrates the effectiveness of ADMIT in empirically mitigating the impact of selection bias.

## 6 Conclusion

In this paper, we propose a novel re-weighting schema to mitigate the impact of selection bias in causal inference with continuous treatments. Under this schema, this paper provides and proves a generalization error bound on the estimation of ADRF based on an IPM distance between the observed (re-weighted) and counterfactual distributions, with theoretical evidence on learning sample weights that alleviate the unreliable counterfactual inference problem caused by selection bias. In the practical implementation, we provide a discretized approximation of the IPM distance with a theoretical guarantee, leading to an algorithm ADMIT that jointly learns re-weighting and inference networks.

## 7 Acknowledgement

This research is supported by National Key R&D Program of China (No. 2021ZD0111700), National Nature Science Foundation of China (No. 62137002, 62176245), Key Research and Development Program of Anhui Province (No. 202104a05020011), Key Science and Technology Special Project of Anhui Province (No. 202103a07020002), Fundamental Research Funds for the Central Universities. We thank the anonymous reviewers for their constructive comments that help improve the manuscript.

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
