# A  Expanded Related Work

## A.1  Causal inference for binary treatments

Much recent work [1, 2, 3, 4] in causal inference focuses on the scenario with binary treatments to estimate causal effects that are defined as the expected difference between the treated and control outcomes, where the selection bias problem has been extensively studied.

In observational data, treatments are typically assigned according to the covariates associated with each unit, resulting in unbalanced covariate distributions among subpopulations that received different treatments, which is known as *selection bias* [5]. It is an important problem on how to alleviate the imbalance which can lead to an unreliable inference. In the binary treatment setting, one approach to the problem of selection bias is re-weighting the units in observational data to balance the treated and control groups [6, 3, 4]. Most of these re-weighing methods are based on the propensity score proposed in [6], which is defined as the probability of treatment assignment conditional on observed covariates. For example, inverse probability of treatment weighting (IPTW) [7] defines a unit's sample weight as the inverse of the probability of receiving the treatment that the unit actually received, and demonstrates that the distribution of covariates in treated and control groups could be balanced using this weight. Hassanpour and Greiner [3] use the importance sampling technique to propose a context-aware weight, which is defined on the basis of the propensity score and emphasizes those units that are important for counterfactual inference. Extended from the standard propensity score, the generalized propensity score (GPS) [8], defined as the conditional density of the treatments conditional on observed covariates, has a similar balancing property in alleviating the selection bias in the continuous treatment setting. Although the methods [9, 10] based on the GPS have some attractive theoretical features, they may suffer from the drawback that the GPS is far more difficult to estimate accurately compared to the standard propensity score [11].

The Integral Probability Metric (IPM) that measures the distance between distributions has been also exploited to mitigate selection bias in some neural network-based methods for causal inference [1, 12, 3, 4]. For example, Shalit et al. [1] propose an algorithm that learns a balanced representation of covariates such that the distributions of treated and control groups look similar, i.e., with reduced IPM distance between these two groups. After that, a linear ridge-regression model is fitted using the factual (observed) distribution on top of learned representations, which bounds the relative error when using the distribution with reverse treatment assignment (counterfactual loss). Unlike regression-based models [1], Li and Fu [12] design a matching estimator based on the learned low-dimensional balanced and nonlinear representations (BNR) for observational data, incorporating a Maximum Mean Discrepancy (MMD) criterion into the model. Yao et al. [2] not only balance the distributions of treated and control groups to reduce selection bias but also preserve the local similarity among units, which provides meaningful constraints on estimating causal effects. However, these methods for adjusting selection bias for binary (also discrete) treatments cannot be easily extended to the continuous treatment settings since there may be uncountably many groups that received different treatments.

## A.2  Connection between causal inference and domain adaptation

Shalit et al. [1] have found a strong connection between causal inference and domain adaptation. Estimating the average treatment effect in the binary treatment setting requires predicting counterfactual outcomes over a different "target" (counterfactual) data distribution based on the "source" (observed) one, which has similarities with domain adaptation methods that focus on transferring knowledge between discrete domains [13, 14, 15]. Shalit et al. [1] employ the IPM distance between treated and control groups to bound the generalization error of estimating causal effects in the binary treatment setting, similar to the generalization bound in domain adaptation given by [16]. In the continuous treatment setting, causal inference is highly related to the continuously indexed domain adaptation [17, 18, 19], which focuses on the scenario where the target domain usually come in a continually evolving manner, such as from day to night. From a domain adaptation perspective, estimating ADRF ($T = t$) requires predicting counterfactual outcomes ($Y^t$) over a continually evolving "target" (counterfactual) distribution $p(X, Y^t | T = s)$ ($s \in [0, 1]$ and $s \neq t$) based on the "source" (observed) distribution $p(X, Y^t | T = t)$. We bound the generalization error of estimating ADRF by an IPM term defined on observed and counterfactual distributions. However, it is impractical to calculate this IPM term since potentially infinite counterfactual distributions may exist in a continuous treatment

scenario. Following [18], we make an assumption that the covariates distributions of subpopulations receiving different treatments smoothly shift, under which we provide a discretized approximation of this IPM term and propose an algorithm to calculate it in practice.

## A.3 Theoretical connection between ADMIT and causal inference for binary treatments

The theoretical part of our work is built on multiple work on causal inference for binary treatments, such as [1, 3, 4]. Shalit et al. [1] prove that expected Precision in Estimation of Heterogeneous Effect (PEHE) loss is upper bounded by the sum of the expected factual loss and expected counterfactual loss when the squared loss is adopted in these two losses. After that, on the basis of the theoretical results related to domain adaptation [13], Shalit et al. [1] bound the counterfactual loss by the factual loss and an IPM, which is adopted in our work. Hassanpour et al. [3] propose context-aware weights that incorporate the valuable context information of each instance, built on top of a representation learning module in [1]. While the context-aware weights are obtained based on the estimation of the propensity score, Johansson et al. [4] propose adaptable sampling weights to balance the treated and control groups, which is adopted in our work.

Our ADRF error upper bound has similarities with generalization bounds in [1, 4], but with significant differences due to the continuity of the treatment. Continuous treatments induce uncountably many potential outcomes per unit, which leads to a more complex selection bias problem than binary treatments. The potentially infinite number of counterfactual distributions is the main challenge since the number of samples for each subpopulation is not enough to estimate the IPM in practice. Therefore, we introduce an assumption to constrain differences in the distributions of subpopulations receiving different treatments. Based on this assumption, we provide the approximation of the IPM term to make it operational and derive an ADRF error upper bound using the IPM term.

# B Proofs

**Theorem 1.** *Let $L$ be the squared loss function, i.e., $L(y, y') = (y - y')^2$. For hypotheses $f_t$ of individual dose-response function $\mu(t, \cdot)$ with marginal loss $\epsilon(f_t) = \mathbb{E}[l_{f_t}(X)]$, there exists a constant $\sigma_{min} \geq 0$, such that,*

$$\text{EMSE}(\mu, \hat{\mu}) \leq \mathbb{E}_T[\epsilon(f_t)] - \sigma_{min}. \tag{1}$$

*Proof.* Let $u$, $v$ be two arbitrary random variables with limited expected values, i.e., $\mathbb{E}[u], \mathbb{E}[u] < \infty$.

Based on the Cauchy–Schwarz inequality, the following inequality holds,

$$(\mathbb{E}[uv])^2 \leq \mathbb{E}[u^2]\mathbb{E}[v^2]. \tag{2}$$

By replacing $u$ and $v$ with $f_t(X) - \mu(t, X)$ and 1 in inequality (2), respectively, we get

$$(\mathbb{E}[f_t(X)] - \mathbb{E}[\mu(t, X)])^2 \leq \mathbb{E}[(f_t(X) - \mu(t, X))^2]. \tag{3}$$

Based on the bias-variance decomposition of the squared loss, the marginal loss $\epsilon(f_t)$ could be decomposed as:

$$\epsilon(f_t) = \mathbb{E}[(Y^t - \mu(t, X))^2] + \mathbb{E}[(f_t(X) - \mu(t, X))^2]. \tag{4}$$

The term $\mathbb{E}[(Y^t - \mu(t, X))^2]$ is a constant determined by the data generation process, denoted by $\sigma_t(Y)$. Combining inequality (3) and equality (4), we get

$$(\hat{\mu}(t) - \mu(t))^2 \leq \epsilon(f_t) - \sigma_t(Y), \tag{5}$$

where $\mu(t) = \mathbb{E}[\mu(t, X)]$ and $\hat{\mu}(t) = \mathbb{E}[f_t(X)]$. Let $\sigma_{min} = min\{\sigma_t(Y)\} \ \forall t \in [0, 1]$, and take expectations on both sides, we have our result.

$\square$

**Lemma 1.** *Let $\mathcal{G}$ be a family of functions $l : \ \mathcal{X} \rightarrow \mathcal{R}$. Assume the per-unit expected loss function $L(f, f') \in \mathcal{G}$ for all $f, f' \in \mathcal{H}$. Then for any $s \in [0, 1]$ and $s \neq t$, we have:*

$$\epsilon(f_t | T = s) \leq \epsilon_w(f_t | T = t) + \text{IPM}_{\mathcal{G}}(p_s, p_t^w). \tag{6}$$

*Proof.* By definitions of the *conditional loss* and $\text{IPM}_{\mathcal{G}}$, the following holds,

$$\epsilon(f_t|T=s) - \epsilon_w(f_t|T=t)$$
$$= \mathbb{E}_{X|T}[l_f(x)|T=s] - \mathbb{E}_{X|T}[w(x)l_f(X)|T=t]$$
$$\leq \left| \int l_f(x)(p_s(x) - p_t^w(x))dx \right|$$
$$\leq sup_{g \in \mathcal{G}} \left| \int g(x)(p_s(x) - p_t^w(x))dx \right|$$
$$= \text{IPM}_{\mathcal{G}}(p_t^w, p_s).$$

**Theorem 2.** *Let* $\text{IPM}_{max} = \max_{s \in [0,1]}\{\text{IPM}_{\mathcal{G}}(p_s, p_t^w)\}$. *The following holds under the conditions of Lemma 1,*

$$\epsilon(f_t) \leq \epsilon_w(f_t|T=t) + \text{IPM}_{max}. \tag{7}$$

*Proof.* By the law of iterated expectation and Lemma 1, we have our result:

$$\epsilon(f_t) = \int \epsilon(f_t|T=s)p(s)ds$$
$$\leq \int (\epsilon_w(f_t|T=t) + \text{IPM}_{max})p(s)ds$$
$$= \epsilon_w(f_t|T=t) + \text{IPM}_{max}.$$

**Assumption 3.** *Let* $p_{t_1}$ *and* $p_{t_2}$ *denote the conditional probability densities of subpopulations that received treatment* $t_1$ *and* $t_2$, *respectively. We assume that there is a constant* $\alpha$ *such that the following inequality holds* $\forall t_1, t_2 \in [0,1]$:

$$\text{IPM}_{\mathcal{G}}(p_{t_1}, p_{t_2}) \leq \alpha |t_1 - t_2|. \tag{8}$$

We bound the difference between the $\text{IPM}_{max}$ and its discretization under Assumption 3 that the probability distributions of subpopulations that received different treatments shift smoothly.

**Lemma 2.** *Suppose we have* $n$ *i.i.d. sample of units, and the* $i$*th unit received a treatment* $t_i \sim p(t)$. *We assume Assumption 3 holds for a constant* $\alpha$. *Then the following holds,*

$$\text{IPM}_{max} \leq \max_{i \in \{1, \cdots, n\}}\{\text{IPM}_{\mathcal{G}}(p_{t_i}, p_t^w)\} + O_p(\frac{\alpha}{\sqrt[3]{n}}). \tag{9}$$

*Proof.* Without loss of generality, assume $t_1 \leq t_2 \leq \cdots \leq t_n$. Consider a sequence of random variables: $\{L_n\}_{n=1,2,\cdots}$, where $L_n = \max_{i \in \{0,1,2,\cdots,n\}} (|t_{i+1} - t_i|)$ ($t_0 = 0$, $t_{n+1} = 1$), we first prove that $L_n$ converges in probability to zero, whose rate of convergence is at least $n^{-1/3}$, i.e., $L_n = O_p(\frac{1}{\sqrt[3]{n}})$.

Let $\beta = p_{max}(t)/p_{min}(t)$, where $p_{min}(t)$ and $p_{max}(t)$ are the minimum and maximum probability of $p(t)$, respectively. Suppose $I_i$ is the interval $[d_{i-1}, d_i)$, where each $I_i$ satisfies $\int_{t \in I_i} p(t)dt = \frac{1}{m}$ ($d_0 = 0$ and $d_m = 1$), for $i = 1, 2 \cdots, m$. Let $A$ denote the event $\exists i \in \{1, \cdots, m\}, \forall j \in \{1, \cdots, n\}, t_j \notin I_i$. Then the following holds,

$$P(L_n \geq \frac{2\beta}{m}) \leq P(A)$$
$$= 1 - \frac{\binom{n-1}{m-1}}{\binom{n+m-1}{m-1}}$$
$$= 1 - \frac{(n-1)!n!}{(n-m)!(n+m-1)!}$$
$$= 1 - \frac{(n-m+1) \times (n-m+2) \times \cdots \times (n-1)}{(n+1) \times (n+2) \times \cdots \times (n+m-1)}$$
$$< 1 - (\frac{n-m}{n})^m.$$

For any $\varsigma > 0$, there exist numbers $1 < M = 2\beta < \infty$ and $N = (\frac{1}{-log(1-\varsigma)})^3$ such that

$$P(L_n \geq \frac{M}{\sqrt[3]{n}}) < 1 - (1 - \frac{\sqrt[3]{n}}{n})^{\sqrt[3]{n}}$$

$$= 1 - (1 - \frac{\sqrt[3]{n}}{n})^{\frac{n}{\sqrt[3]{n}} \times \frac{(\sqrt[3]{n})^2}{n}}$$

$$\simeq 1 - e^{-\frac{1}{\sqrt[3]{n}}}$$

$$< \varsigma$$

for any $n > N$. Therefore, $L_n = O_p(\frac{1}{\sqrt[3]{n}})$. Under Assumption 3, $\forall i \in \{0, 1, \cdots, n, n+1\}$, $\forall s \in [t_i, t_{i+1}]$, the following holds,

$$\text{IPM}_{\mathcal{G}}(p_s, p_t^w) \leq \text{IPM}_{\mathcal{G}}(p_{t_i}, p_t^w) + \text{IPM}_{\mathcal{G}}(p_{t_i}, p_s)$$

$$\leq \text{IPM}_{\mathcal{G}}(p_{t_i}, p_t^w) + O_p(\frac{\alpha}{\sqrt[3]{n}}).$$

By the definition of $\text{IPM}_{max}$, we have our result. $\square$

**Lemma 3.** *Let $p_{\Delta s} = P_{X|T}(x|t \in [s, s+\delta])$ $(0 < \delta < 1)$ denote the conditional density of covariates when $t \in [s, s+\delta]$. Then the following holds under Assumption 3,*

$$\text{IPM}_{\mathcal{G}}(p_s, p_t^w) \leq \text{IPM}_{\mathcal{G}}(p_{\Delta s}, p_t^w) + \alpha\delta. \tag{10}$$

*Proof.* Due to the triangle inequality for the *Integral Probability Metric*,

$$\text{IPM}_{\mathcal{G}}(p_s, p_t^w) \leq \text{IPM}_{\mathcal{G}}(p_{\Delta s}, p_t^w) + \text{IPM}_{\mathcal{G}}(p_{\Delta s}, p_s).$$

By the definition of the $\text{IPM}_{\mathcal{G}}$, the following holds,

$$\text{IPM}_{\mathcal{G}}(p_{\Delta s}, p_s)$$

$$= sup_{l \in \mathcal{G}} \left| \int_x l(x)p_s(x)d(x) - \int_s^{s+\delta} p(t|t \in [s, s+\delta t])dt \int_x l(x)p_t(x)dx \right|$$

$$= sup_{l \in \mathcal{G}} \left| \int_s^{s+\delta t} p(t|t \in [s, s+\delta])dt \int_x l(x)(p_s(x) - p_t(x))dx \right|$$

$$\leq sup_{l \in \mathcal{G}} \int_s^{s+\delta t} p(t|t \in [s, s+\delta])dt \left| \int_x l(x)(p_s(x) - p_t(x))dx \right|$$

$$\leq \alpha\delta.$$

Therefore, we have our result. $\square$

**Theorem 3.** *Suppose we have $n$ i.i.d. sample of units, and the $i$th unit received a treatment $t_i$. Let $\text{IPM}_{\Delta max} = \max_{i \in \{1, \cdots, n\}} \{\text{IPM}_{\mathcal{G}}(p_{\Delta t_i}, p_t^w)\}$. We assume Assumption 3 holds for a constant $\alpha$. Then, for a neighborhood size $0 < \delta < 1$ we have,*

$$\epsilon(f_t) \leq \epsilon_w(f_t|T = t) + \text{IPM}_{\Delta max} + O_p(\frac{\alpha}{\sqrt[3]{n}}) + \alpha\delta. \tag{11}$$

*Proof.* Following Lemma 2 and Lemma 3, we could proof Theorem 3. $\square$

**Property 1.** *The minimum $\alpha$ that meets the conditions of Assumption 3 is*

$$\alpha_{min} = \max_{s \in [0,1]} \{\lim_{\delta \to 0} \frac{IPM(p_s, p_{s+\delta})}{\delta}\}. \tag{12}$$

*Proof.* Let $t_1 < t_2$ denote two arbitrary variables in $[0, 1]$. We divide $[t_1, t_2]$ into $n$ intervals, each of length $\eta = \frac{t_2 - t_1}{n}$.

According to the triangle inequality for the Integral Probability Metric, we can get,

$$\text{IPM}(t_1, t_2) \leq \sum_{i=0}^{n-1} \text{IPM}(t_1 + \eta i, t_1 + \eta(i+1)).$$

Let $\alpha(t)$ denote $\lim_{\delta \to 0} \frac{IPM(p_s, p_{s+\delta})}{\delta}$, then we have,

$$\int_{t_1}^{t_2} \alpha(t)dt = \lim_{n \to \infty} \sum_{i=0}^{n-1} \text{IPM}(t_1 + \eta i, t_1 + \eta(i+1)).$$

Therefore, $\forall t_1, t_2 \in [0, 1]$,

$$\frac{\text{IPM}(t_1, t_2)}{t_2 - t_1} \leq \alpha_{min}.$$

On the other hand, we can get $\alpha \geq \alpha_{min}$ according to the definition of $\alpha$. In other words, $\alpha_{min}$ is the minimum $\alpha$ that meets the conditions of Assumption 3. □

### B.1 Generalization bound based on finite samples

In this section, we refer to a lemma from [20] to give the finite sample guarantee of Theorem 3.

**Lemma 4.** (Sriperumbudur et al. [20]) *Let $\mathcal{X}$ be a measureable space. Suppose $k$ is a universal, measurable kernel such that $\sup_{x \in \mathcal{X}} k(x, x) \leq C$ and $\mathcal{H}$ the reproducing kernel Hilbert space induced by $k$, with $v := \sup_{x \in \mathcal{X}, f \in \mathcal{H}} \leq \infty$. Then, with $\hat{p}, \hat{q}$ the empirical distributions of $p$, $q$ from $m$ and $n$ samples, and with probability at least $1 - \xi$, we have,*

$$|\text{IPM}_{\mathcal{H}}(p, q) - \text{IPM}_{\mathcal{H}}(\hat{p}, \hat{q})| \leq \sqrt{18 v^2 log\frac{4}{\xi}} \left( \frac{1}{\sqrt{m}} + \frac{1}{\sqrt{n}} \right). \tag{13}$$

With Lemma 4 and Theorem 3, we can give the finite sample guarantee for the proposed algorithm ADMIT.

**Theorem 4.** *Suppose we have $n$ i.i.d. sample of units with an empirical measure $\hat{p}$, and the $i$th unit received a treatment $s_i$. Let $n_s$ denote the number of units belonging to $[s, s + \delta]$, and $\widehat{\text{IPM}}_{\Delta max} = \max_{i \in \{1, \cdots, n\}} \{\text{IPM}_{\mathcal{G}}(\hat{p}_{\Delta s_i}, \hat{p}_{\Delta t}^w)\}$. We assume Assumption 3 holds for a constant $\alpha$. Then, for a neighborhood size $0 < \delta < 1$, we have,*

$$\epsilon(f_t) \leq \epsilon_w(f_t | T = t) + \widehat{\text{IPM}}_{\Delta max} + \sqrt{18 v^2 log\frac{4}{\xi}} D_{n_s} + \sigma_{Y_t} + O_p(\frac{\alpha}{\sqrt[3]{n}}) + \alpha\delta, \tag{14}$$

*where $D_{n_s} = \max_{i \in \{1, \cdots, n\}} \{\frac{1}{\sqrt{n_{s_i}}} + \frac{1}{\sqrt{n_t}}\}$.*

## C Experimental Details

### C.1 Dataset descriptions

**News.** The News dataset, consisting of a random sample of 5,000 news items from the NY Times corpus [21], was originally introduced as a benchmark for counterfactual inference in the binary treatment setting [22]. For each news item $\boldsymbol{x}$, the $i$th dimension $x_i$ represents the number of occurrences of the $i$th word. Following [22, 23], to give meaning to our treatments and outcomes, we let treatment $T$ and outcome $Y^T$ represent the time readers spending on the news and their satisfaction with the news, respectively. The same version of the News dataset as DRNet (https://github.com/d909b/drnet) is used in this work.

**TCGA.** The TCGA project collected gene expression data for various types of cancer from 9,659 individuals, from which we select the 4,000 most variable genes as features to build our dataset as in [23]. We scaled the features of each patient to have norm 1. To give meaning to our treatments

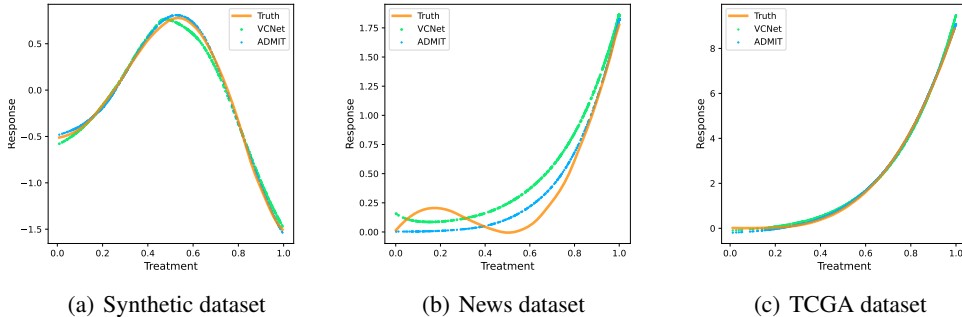

|                  | (a) Synthetic dataset | (b) News dataset | (c) TCGA dataset |

Figure 1: Estimated ADRF on testing set from a typical run of ADMIT and VCNet. The truth is shown in solid orange line.

Table 1: Summary description of datasets.

| Dataset            | Simulation | News  | TCGA  |
|--------------------|-----------|-------|-------|
| Number of samples  | 5,000     | 5,000 | 9,659 |
| Number of features | 6         | 3,477 | 4,000 |

and outcomes, we let treatment $T$ and outcome $Y^T$ represent the medication dosage and the risk of cancer recurrence after receiving corresponding treatment, respectively. The same version of the TCGA dataset as SCIGAN (`https://github.com/ioanabica/SCIGAN`) is used in this work.

A summary description of the datasets is shown in Table 1. We randomly split each dataset into training set (67%), validation set (23%), and test set (10%). The validation dataset is used for hyperparameter selection.

## C.2   Implement details

**Baselines.** We implement entropy balancing for continuous treatments (EBCT) [24] using `https://github.com/EddieYang211/ebal-py`, and GPS using Python package "causal-curve" [25] `https://github.com/ronikobrosly/causal-curve`. Moreover, we use the publicly available implementation of SCIGAN provided by [23]: `https://github.com/ioanabica/SCIGAN`, and implementations of VCNet and DRNet provided by [26]: `https://github.com/lushleaf/varying-coefficient-net-with-functional-tr`. We implement our model on PyTorch with an Nvidia RTX3090 GPU. The implementation of the varying coefficient prediction head we use to build the inference and re-weighing networks is based on [26], and the kernel we apply in calculating MMD is the Gaussian kernel based on `https://github.com/oddrose/cfrnet`.

**Parameter setting.** We tune parameters based on the validation split of each dataset, and use the EMSE for evaluation. We tune the following parameters: network learning rate $lr \in \{0.005, 0.001, 0.0005, 0.0003, 0.0001\}$, batch size $bs \in \{100, 200, 500, 1,000\}$, and neighbourhood size $\delta \in \{0.05, 0.1, 0.2\}$. All networks are trained for 200 epochs during tuning.

## C.3   Dose-response curve

To observe the effectveness of our model visually, the estimated dose-response curves of ADMIT and VCNet and the truth are plotted in Figure 1. Across different datasets, when the true ADRF is simpler, both ADMIT and VCNet fit better. Moreover, ADMIT always be able to fit the ADRF better than VCNet, especially when the true ADRF is relatively complex.