# OpenReview forum: "Generalization Bounds for Estimating Causal Effects of Continuous Treatments"
_NeurIPS.cc/2022/Conference — NeurIPS 2022 Accept_

### Official Review · Reviewer_TaAx · 2022-07-08

**Rating:** 7
**Confidence:** 4
**Soundness:** 3 good
**Presentation:** 3 good
**Contribution:** 3 good

**Summary:**

The authors tackle the problem of estimating causal effects under continuous treatments by proposing a novel framework to estimate the average dose response function (ADRF). The core idea of their framework is to minimize a bound on the ADRF loss (instead of minimizing the error in estimating the ADRF directly). The ADRF loss consists of both a factual loss (exact) and an upper bound on the counterfactual loss – which is derived via a re-weighting schema  and the use of an Integral Probability Metric (IPM) that measures the distance between the factual and counterfactual distributions.

The authors provide a practical implementation of this theoretical bound under a smoothness assumption on the shift between covariate distributions in different subpopulations. They call their algorithm ADMIT, which they test in synthetic and semi-synthetic settings.


**Questions:**

It would be nice for the authors to give more intuition for how strong assumption 3 is? Seeing as this is a core assumption for the approximation of the IPM term, the authors need to provide a little bit more discussion surrounding it.

Minor typos/spelling errors:

Line 127: $\hat{u}(t)$ should be $\hat{\mu}(t)$

Line 173: What’s worse is that

Line 16/Line 119: A transition word other than “besides” may be more appropriate


**Limitations:**


No, there was not a significant discussion of the limitations of the approach, which is a weakness of this paper. See above for suggestions to improve this (i.e. assumption 3 discussion, more thorough empirical analyses).


**Strengths And Weaknesses:**


Strengths:

[quality + significance] - The authors provide a clear and well-motivated development of their upper bound on the ADRF loss. Their connection of the theory to practical implementation was also particularly well done.

Weaknesses:

[quality] - There are a few improvements that could be made to the empirical results section of the paper. Increasing the thoroughness of the empirical section is the only major weakness in my mind. For one, it would be useful to provide another view of the results by e.g. providing the actual dose response curve (i.e. response vs dosage for the estimation and the ground truth). This would help the reader get a better idea as to which regions of the treatment may be better (in terms of estimation) vs not. Secondly, it would be interesting to see how the method performs in the setting where some of the confounders were unobserved (or simulated to be unobserved, i.e. you conceal them). Although an initial assumption is one of unconfoundedness, characterizing the failure modes of the method is generally useful. At a high level, the authors need to do a better job at discussing the limitations and failure modes of their approach.

[clarity] (minor) - There are a few areas where there are some grammatical and spelling errors. See below for a few examples (not exhaustive).

---

> ### Author Response · Authors · 2022-08-02
> **Response to Reviewer TaAx**
>
> **Comment1:** There are a few improvements that could be made to the empirical results section of the paper. Increasing the thoroughness of the empirical section is the only major weakness in my mind. For one, it would be useful to provide another view of the results by e.g. providing the actual dose response curve (i.e. response vs dosage for the estimation and the ground truth). This would help the reader get a better idea as to which regions of the treatment may be better (in terms of estimation) vs not.
>
> **Response 1:** Many thanks for your constructive suggestion. We compare the estimated dose-response curves of ADMIT and VCNet with the truth in a new subsection (C.3) added in the revised appendix. To observe the effect of our model visually, the estimated dose-response curves of ADMIT and VCNet and the truth are plotted in Figure 1 in the revised appendix. Across different datasets, when the true ADRF is simpler,  both ADMIT and VCNet fit better. Moreover, ADMIT always be able to fit the ADRF better than VCNet, especially when the true ADRF is relatively complex (see Figure 1 in the appendix).
>
> **Comment2:** Secondly, it would be interesting to see how the method performs in the setting where some of the confounders were unobserved (or simulated to be unobserved, i.e. you conceal them). Although an initial assumption is one of unconfoundedness, characterizing the failure modes of the method is generally useful. At a high level, the authors need to do a better job at discussing the limitations and failure modes of their approach.
>
> **Response 2:** Many thanks for your kind reminder. The ADMIT will fail in the setting where some of the confounders are unobserved since the conditional causal effect is unidentifiable, i.e., $\mathbb E[Y^t|x]\neq\mathbb E[Y|x,t]$. It is an interesting research topic to extend the theoretical results of ADMIT to scenarios where the unconfoundedness assumption is not valid, e.g., introducing the instrumental variables.
>
> **Question 1:** It would be nice for the authors to give more intuition for how strong assumption 3 is? Seeing as this is a core assumption for the approximation of the IPM term, the authors need to provide a little bit more discussion surrounding it.
>
> **Answer 1:** Thanks for your careful review. It is easy to find a constant $\alpha$ that satisfies Assumption 3 when the output of the hypothesis $f_t$ is finite, e.g., survival years after taking a certain medicine, which is reasonable in applications. The main concern is that a large $\alpha$ may lead to a loose bound, but it could be illustrated that this is not a common scenario. On the one hand, the RHS of Equation (17) represents the bound of the worst case. Without loss of generality, assume $s_1\le s_2\le \cdots\le s_n$, and let $IPM(p_{s_i},p_{s_{i+1}})=\alpha_i|s_{i+1}-s_i|$. It is not difficult to prove that $\alpha$ defined in Assumption 3 satisfies $\alpha=max_{s\in[0,1]}\{\lim_{\delta\to0}\}\frac{IPM(p_s,p_{s+\delta})}{\delta}\ge max_{i\in\{1,2,\cdots,n-1\}}(\alpha_i)$ according to the triangle inequality for the Integral Probability Metric. However, during the proof of Lemmata 2 and 3, all $\alpha_i$ are enlarged to $\alpha$, e.g., the inequality $IPM_\mathcal{G}(p_{s_i}, p_t^w)+IPM_\mathcal{G}(p_{s_i}, p_s)\le IPM_\mathcal{G}(p_{s_i}, p_t^w) + O_p(\frac \alpha {\sqrt[3] n})$ in line 106 of the appendix. This is necessary for the proof, but it also shows that the RHS of Equation (17) represents the bound of the worst case. On the other hand, the bound will be loose when $\alpha$ is large and $\forall i, \alpha_i=\alpha$, i.e., the worst case happens. Intuitively, it is not common for each $\alpha_i$ to be large. For instance, the dose of a particular medicine may depend on the age of the patient, and $\forall i, \alpha_i=\alpha$ means that the age distributions of groups taking similar doses vary considerably, which is unreasonable.
>
> **Comment3:**
>
> Minor typos/spelling errors:
>
> Line 127: $\hat u{(t)}$ should be $\hat \mu{(t)}$
>
> Line 173: What’s worse is that
>
> Line 16/Line 119: A transition word other than “besides” may be more appropriate
>
> **Response 2:** Thanks for your careful review. We have thoroughly checked and corrected the grammatical errors and typos we found in the revised manuscript.

---

> > ### Comment · Reviewer_TaAx · 2022-08-08
> > **Thanks for the response.**
> >
> > Thank you for clarifying my concerns. I will be maintaining my score.

---

### Official Review · Reviewer_XSmM · 2022-07-09

**Rating:** 3
**Confidence:** 3
**Soundness:** 2 fair
**Presentation:** 2 fair
**Contribution:** 3 good

**Summary:**

This paper derives an error bound for treatment effect estimation in the continuous treatment (dosage) regime. The key insight is to relate this error to the bias introduced by distributional shift in treatment assignment using IPM distances between distributions. The authors further provide an algorithm inspired by the theoretical upper bound as well as empirical validation for the proposed method.



**Questions:**

Extracted from the above section:

* Lemma 1 seems somewhat restrictive given that the loss function has to be in the family of functions that defines the IPM. What are the implications for the squared loss?
* EMSE is a population quantity. What are the finite sample guarantees (if any)?
* How do you actually calculate the IPM gradients and how does it contribute to the computational complexity?
* How do the sample and computational complexities depend on the choice of $\delta$?
*  What were the computational bottlenecks of ADMIT? How does the GPU usage compare across methods?
* Given the minor DGP modifications, have you tuned the parameters of the baselines (e.g. VCNet) to the new DGP?

**Limitations:**

The authors have not addressed the computational limitations of their algorithm besides the fact that $\simeq 3000$ GPUs were used.

**Strengths And Weaknesses:**

The paper is by-and-large an extension of [1] to continuous treatments domains by leveraging a discretized version of the IPM metric used in [1], as well as importance sampling with learned weights ([2], [3]). The idea itself is a good addition to the causal inference literature, but I have reservations regarding the execution, both from a technical and quality of writing standpoint.

Overall structure:
* There is not enough contextualization with previous work. For example, [1], [2], [3] are only briefly mentioned and their connection with the current work is not made obvious. The entire *Related Work* section lacks clarity.
* The introduction is also confusing, especially for those unfamiliar with the advances in this specific area of causal inference. The different paragraphs don't seem connected and it's unclear what the state of the art that this paper is improving upon actually is.
* Section 4.4 and the Algorithm Box are too underdeveloped. It it unclear what the different quantities ($\phi, h, w$) until you read the text and the text doesn't have enough explanations about how to execute the different steps of the algorithm (e.g. how to computer the IMP gradient or how to choose $\delta$ appropriately). The appendix is very sparse and doesn't contain additional information that could answer this question.

Technical weaknesses:
* Theorem 1 also holds for $\sigma_{min}=0$, for example when $Y^t=f_t(x)$, i.e. the counterfactual outcome under treatment $t$ depends solely on observed features.
* Lemma 1 seems somewhat restrictive given that the loss function has to be in the family of functions that defines the IPM. What are the implications for the squared loss?
* EMSE is a population quantity. What are the finite sample guarantees (if any)?
* How do you actually calculate the IPM gradients and how does it contribute to the computational complexity? How do the sample and computational complexities depend on the choice of $\delta$?
* Concerns about reproducibility: the simulations were run on 3080 GPUs according to the appendix. What were the computational bottlenecks? How does the GPU usage compare across methods?
* The experimental results are encouraging, but there are some issues with the performance of the benchmarks. For example, the DGP is similar to the one in the VCNet paper [4], but their performance were more along $\simeq 0.15$, rather than the $\simeq 0.19$ found in this paper. And if the discrepancy comes from the minor DGP modifications, have you tuned the parameters of the VCNet to the new DGP?

Overall, I don't think this work is ready for publication yet.

[1] Uri Shalit, Fredrik D Johansson, and David Sontag. Estimating individual treatment effect: Generalization bounds and algorithms.

[2] Negar Hassanpour and Russell Greiner. Counterfactual regression with importance sampling weights.

[3] Fredrik D Johansson, Uri Shalit, Nathan Kallus, and David Sontag. Generalization bounds and
360 representation learning for estimation of potential outcomes and causal effects.

[4] Lizhen Nie, Mao Ye, Dan Nicolae, et al. VCNet and functional targeted regularization for learning causal effects of continuous treatments.

---

> ### Author Response · Authors · 2022-08-02
> **Response to Reviewer XSmM: Part1**
>
> **It should be clarified that our experiments were run on a machine with one Nvidia RTX3090 GPU, not on a machine with more than 3000 GPUs.** We apologize for the vague statements in the appendix that mislead your understanding.
>
> **Question 1:** Lemma 1 seems somewhat restrictive given that the loss function has to be in the family of functions that defines the IPM. What are the implications for the squared loss?
>
> **Answer 1:** There are two reasons for applying the squared loss. Firstly, the squared loss is commonly used in regression, which coincides with the casual inference that estimating the potential outcome is a regression problem since $y$ is continuous. Secondly, when using the squared loss $L(y,y')=(y-y')^2$ and letting $l_{f_t}( {x}):=\mathbb{E}_{Y^t|{X}}[L(Y^t,f_t({X}))| {X}= {x}]$ belong to the $\mathcal G$ in Lemma, the IPM discrepancy is a distance, which implies that minimizing the discrepancy to zero guarantees balancing covariates distributions. Gretton et al. [1] prove that the discrepancy is a distance, i.e., if $IPM_\mathcal G(p,q)=0$, then $p=q$, when $L$ is the squared loss, and $f_t\in \mathcal H$, a subset of the reproducing kernel Hilbert space (RKHS).
>
> **Question 2:** EMSE is a population quantity. What are the finite sample guarantees (if any)?
>
> **Answer 2:** Thanks for your kind reminder. We refer to a lemma from [2] to give the finite sample guarantee of Theorem 3.
>
> Suppose we have $n$ i.i.d sample of units with an empirical measure $\hat p$, and the $i$th unit received a treatment $s_i$. Let $n_s$  denote the number of units belonging to $[s, s+\delta]$, and $I\hat{P}M_{\Delta max}=max_{i\in\{1, \cdots, n\}}(IPM_\mathcal{G}(\hat p_{\Delta s_i}, \hat p_{\Delta t}^w))$. We assume Assumption 3 holds for a constant $\alpha$. Then, for a neighborhood size $0<\delta<1$ we have,
> $$
> \epsilon(f_t)\leq \epsilon_w(f_t|T=t)+I\hat{P}M_{\Delta max}+\sqrt{18v^2log\frac{4}{\xi}}D_{n_s}+O_p(\frac \alpha {\sqrt[3] n})+\alpha\delta,
> $$
> where $D_{n_s}=max_{i\in\{1, \cdots, n\}}\{(\frac {1}{\sqrt{n_{s_i}}}+ \frac {1}{\sqrt{n_t}})\}$.
>
> **Question 3:** How do you actually calculate the IPM gradients and how does it contribute to the computational complexity?
>
> **Answer 3:** From a practical point of view, the IPM gradients are calculated by using PyTorch’s automatic differentiation engine that powers neural network training.
>
> From a theoretical point of view, the IPM gradients are calculated as follows. Given a neural network $G_{\theta_w,\theta_\phi}$, where $\theta_w$, $\theta_\phi$ represent the parameters of re-weighting and representation network, respectively. Let $U=(u_1, u_2,\cdots u_m)$ denote the inputs drawn from $p_{\Delta l}^w$ in line 6 of Algorithm 1, let $V=(v_1, v_2,\cdots v_n)$ denote the inputs drawn from $p_{\Delta k}$, and let $Z_\theta=(z_1,z_2,\cdots,z_m)$ with $z_i=G_\theta(u_i)$ and $\theta=(\theta_w, \theta_\phi)$. As explained in the paper, IPM becomes the Maximum Mean Discrepancy (MMD) metric when we choose a family of norm-1 reproducing kernel Hilbert space (RKHS) functions. Given a differentiable kernel $k$, we minimize $IPM(p_{\Delta l}^w,p_{\Delta k})=C(Z_\theta, V)$ as a function of $\theta$, where
> $$
> C(Z_\theta, V)=\frac{1}{n^2}\sum_{i=1}^n\sum_{j=1}^n k({z_i},{z_j})-\frac{2}{mn}\sum_{i=1}^n\sum_{j=1}^m k({z_i},{v_j})+\frac{1}{m^2}\sum_{i=1}^m\sum_{i=j}^m k({v_i},{v_j}).
> $$
> Then, the IPM gradients could be calculated with the chain rule as follows:
> $$
> \Delta_\theta C(Z_\theta, V)=\frac1n\sum_{i=1}^n\sum_{j=1}^m\frac{\partial C(Z_\theta, V)}{\partial z_j}\frac{\partial G_\theta(u_i)}{\partial \theta}.
> $$
> The additional complexity introduced by the $IPM$ term is about $O(\eta n^2 d)$, where $\eta=\lceil 1/\delta\rceil$, $n$ denotes the sample size, and $d$  denotes the dimension of covariates $x$. Moreover, the spending time comparison between our model and the compared models with the same settings (using one Nvidia **RTX3090** GPU) is given in **Answer 5**. It could be seen that using the $IPM$ term increases the complexity but is within acceptable limits.
>
> **Question 4:** How do the sample and computational complexities depend on the choice of $\delta$?
>
> **Answer 4:** The computational complexity is about $O(\eta n^2 d)$, where $\eta=\lceil 1/\delta\rceil$, $n$ denotes the sample size, and $d$  denotes the dimension of covariates $x$. Specifically, we list the times when ADMIT runs an epoch (a total of 200 epochs are trained in our experiment) on the synthetic dataset with $\delta\in(0.05,0.1,0.2,0.25,0.5)$. The results show that the time spent is acceptable.
>
>   | $\delta$           | 0.5   | 0.25  | 0.2   | 0.1   | 0.05  |
>   | ------------------ | ----- | ----- | ----- | ----- | ----- |
>   | **Time** (seconds) | 0.196 | 0.286 | 0.422 | 0.891 | 2.498 |

---

> > ### Author Response · Authors · 2022-08-02
> > **Response to Reviewer XSmM: Part2**
> >
> > **Question 5:**  What were the computational bottlenecks of ADMIT? How does the GPU usage compare across methods?
> >
> > **Answer 5:** We apologize for the vague statements in the paper that led a misunderstanding that ADMIT required more than 3000 GPUs. In fact, ADMIT requires only one Nvidia **RTX3090** GPU. We list the times when ADMIT, VCNet, and DRNet run an epoch, and we can see that ADMIT does not have significant computational bottlenecks.
> >
> >   | model               | ADMIT | VCNet | DRNet |
> >   | ------------------- | ----- | ----- | ----- |
> >   | **Time**  (seconds) | 0.422 | 0.174 | 0.229 |
> >
> > **Question 6:**  Given the minor DGP modifications, have you tuned the parameters of the baselines (e.g. VCNet) to the new DGP?
> >
> > **Answer 6:** Yes, parameters have been tuned for the baselines. The slight difference in the performance of VCNet (0.15 vs 0.19) you mentioned should come from two sources. On the one hand, as you mentioned, there is some difference in the data generation process (DGP). On the other hand, The [News](https://www.fredjo.com/files/NEWS_csv.zip) dataset contains 50 realisations of a stochastic outcome model, and each realisation consists of 5000 randomly sampled news. We get the results by randomly selecting a certain realisation and repeating it several times. This may be different from the VCNet setup, which is not mentioned in the paper or [code](https://github.com/lushleaf/varying-coefficient-net-with-functional-tr) of VCNet.
> >
> > **Comment1:** There is not enough contextualization with previous work. For example, [1], [2], [3] are only briefly mentioned and their connection with the current work is not made obvious. The entire *Related Work* section lacks clarity.
> >
> > **Response 1:** Thanks for your kind reminder. The related studies have been introduced in detail in a new subsection (A.3) in Expanded Related Work in the revised appendix. In addition, we also explain the connection between our work and these efforts.
> >
> > **Comment2:** The introduction is also confusing, especially for those unfamiliar with the advances in this specific area of causal inference. The different paragraphs don't seem connected and it's unclear what the state of the art that this paper is improving upon actually is.
> >
> > **Response 2:** Unlike binary treatments, causal inference with continuous treatments is largely understudied. We introduce three state-of-the-art works, VCNet, DRNet and SCIGAN, in causal inference with continuous treatments in the introduction. However, these works could not well mitigate selection bias in the continuous setting. This paper achieves the state of the art by deriving an ADRF error upper bound, which provides theoretical guarantees to mitigate selection bias among a theoretically infinite number of subgroups.
> >
> > **Comment3:** Section 4.4 and the Algorithm Box are too underdeveloped. It it unclear what the different quantities (ϕ,h,w) until you read the text and the text doesn't have enough explanations about how to execute the different steps of the algorithm (e.g. how to computer the IMP gradient or how to choose δ appropriately). The appendix is very sparse and doesn't contain additional information that could answer this question.
> >
> > **Response 3:** Thanks for your careful review. We explain the calculation process of the IMP gradient in **Answer 3**.
> >
> > **Comment4:** Theorem 1 also holds for $σ_{min}=0$, for example when $Y^t=f_t(x)$, i.e. the counterfactual outcome under treatment t depends solely on observed features.
> >
> > **Response 4:** Thanks for your kind reminder. We have made a correction in the revised manuscript.
> >
> > [1] Corinna Cortes and Mehryar Mohri. Domain adaptation and sample bias correction theory and algorithm for regression. Theoretical Computer Science, 519:103–126, 2014.
> >
> > [2] Corinna Cortes, Yishay Mansour, and Mehryar Mohri. Learning bounds for importance weighting. Advances in neural information processing systems, 23, 2010.
> >
> > [3] Bharath K Sriperumbudur, Kenji Fukumizu, Arthur Gretton, Bernhard Schölkopf, and Gert RG Lanckriet. On integral probability metrics,\phi-divergences and binary classification. arXiv preprint arXiv:0901.2698, 2009. Arthur Gretton, Karsten M Borgwardt, Malte J Rasch, Bernhard Schölkopf, and Alexander Smola. A kernel two-sample test. The Journal of Machine Learning Research, 13(1):723–773, 2012.

---

> ### Author Response · Authors · 2022-08-08
> **Could you re-evaluate our paper based on our revision and response? Thank you.**
>
> Since the stage of the Reviewer-Author discussion period is closing soon, we hope the reviewer can re-evaluate our paper based on our updated paper and detailed response, especially the clarification of the computational complexity.

---

> > ### Comment · Reviewer_XSmM · 2022-08-08
> > **Rebuttal Response**
> >
> > I thank the authors for the detailed answers. I re-read the paper and, while the author responses make the context and technical contribution more clear, I believe it should not take a 2 page response to get the main points across. All of these should be included in the paper in some form. I want to be clear: I think that the work is technically strong and has a good amount of impact, but the exposition obscures many of these strengths. Because of this, I maintain my original assessment of the work and I will be keeping my score unchanged.

---

### Official Review · Reviewer_WSLo · 2022-07-11

**Rating:** 5
**Confidence:** 5
**Soundness:** 2 fair
**Presentation:** 3 good
**Contribution:** 2 fair

**Summary:**

This paper estimates the causal effects of continuous treatments. To balance the covariates among infinite subpopulations, they learn re-sampling weights that reduce the IPM distance between observed and counterfactual groups. Thus, they derive an upper bound on the estimated counterfactual error and demonstrate experimentally the proposed algorithm ADMIT based on the derived upper bound outperforms GPS, EBCT, DRNet, SCIGAN, and VCNet.

**Questions:**

This work provides a discretized approximation of the IPM term.  Since there are limited samples while the number of domains is infinite, to overcome this challenge, they bound the difference between the IPM and its discretization under the assumption 3 that the probability distributions of subpopulations that received different treatments shift smoothly. However, how could this model handle a substantial shift and give a continuous and complete dose-response curve?

**Limitations:**

The experimental analysis is relatively insufficient for the continuous treatments.

**Strengths And Weaknesses:**

Strengths:
1. provide a theoretical guarantee to the causal effect estimation of continuous treatments.
2. give a comprehensive summary of the current studies on the continuous treatment effect estimation

Weaknesses:
1. lack of innovation. The proposed framework is mostly based on previous works such as GPS, CFR, and DRNet.
2. limited contribution. although the author provides the generalization bounds for estimating causal effects of continuous treatment, the theoretical proofs are the extension of CFR [Shalit, Uri, Fredrik D. Johansson, and David Sontag. "Estimating individual treatment effect: generalization bounds and algorithms." International Conference on Machine Learning. PMLR, 2017.]

---

> ### Author Response · Authors · 2022-08-02
> **Response to Reviewer WSLo**
>
> **Comment1:** lack of innovation. The proposed framework is mostly based on previous works such as GPS, CFR, and DRNet. limited contribution. although the author provides the generalization bounds for estimating causal effects of continuous treatment, the theoretical proofs are the extension of CFR [1]
>
> **Response 1:** On the basis of previous studies, we extend the generalization bounds for estimating causal effects of binary treatments to the generalization bound of continuous treatments. However, we claim that this extension is nontrivial and meaningful. Firstly, it's an important research area to estimate the causal effect of continuous treatments because continuous treatments arise in many fields, including economics and medicine. Secondly, causal inference with continuous treatments is largely understudied and far more challenging than binary treatments. Continuous treatments induce uncountably many potential outcomes per unit, which leads to a more complex selection bias problem than binary treatments. Therefore, it is nontrivial to extend the causal inference of binary treatments to continuous treatments. Finally, it is not so straightforward and simple to extend the causal inference of binary treatments to continuous treatments. Due to the potentially infinite number of covariates distributions, it is hard to mitigate the selection bias problem in the setting of continuous treatments via the theories and technologies in causal inference for binary treatments. We introduce Assumption 3 to constrain differences in the distributions of subpopulations receiving different treatments. Based on Assumption 3, we provide the approximation of the IPM term and related theoretically support to make it operational.
>
> On the other hand, it may not be a good criterion for evaluating a study by whether it is an extension of previous works. For example, CFR [1] is a remarkable work on causal inference. However, CFR was also built on the basis of literature [2, 3] in domain adaptation.
>
> **Question 1:** This work provides a discretized approximation of the IPM term. Since there are limited samples while the number of domains is infinite, to overcome this challenge, they bound the difference between the IPM and its discretization under the assumption 3 that the probability distributions of subpopulations that received different treatments shift smoothly. However, how could this model handle a substantial shift and give a continuous and complete dose-response curve?
>
> **Answer 1:** Intuitively, a small $\alpha$ in Assumption 3 indicates a smooth covariates shift, i.e., a slight selection bias. At first, when the shift among different subpopulations is smooth (a slight selection bias with $\alpha=2$), Table 1 indicates that ADMIT outperforms the baselines. In addition, Figure 3 also shows that ADMIT has consistent performance and outperforms the baselines when the selection bias gradually becomes more intense (increase $\alpha$ from 1 to 8), which indicates that ADMIT can handle the case whose shift is slightly sharp. When the shift is too substantial, ADMIT and other baselines may not be able to infer well. Note that a substantial shift does not seem to be common in real-world applications. For instance, the dose of a particular medicine may depend on the age of the patient, and then a substantial shift means that the age distributions of groups taking similar doses vary considerably, which is unreasonable. Besides, our extensive experiments show that ADMIT could give a continuous and complete dose-response curve. For more details, please refer to Section C.3 in the appendix.
>
> [1] Shalit, Uri, Fredrik D. Johansson, and David Sontag. "Estimating individual treatment effect: generalization bounds and algorithms." International Conference on Machine Learning. PMLR, 2017.
>
> [2] Corinna Cortes and Mehryar Mohri. Domain adaptation and sample bias correction theory and algorithm for regression. Theoretical Computer Science, 519:103–126, 2014.
>
> [3] Mansour, Yishay, Mohri, Mehryar, and Rostamizadeh, Afshin. Domain adaptation: Learning bounds and algorithms. 2009.

---

### Official Review · Reviewer_hMRr · 2022-07-12

**Rating:** 6
**Confidence:** 4
**Soundness:** 3 good
**Presentation:** 4 excellent
**Contribution:** 3 good

**Summary:**

This work derives a new generalization bound for estimating causal effects of continuous treatments. The new bound is based on the idea of generalized propensity score reweighting, and is obtained via a sequence of inequalities, translating the original marginal treatment effect which is hard to estimate to a new quantity that is easier to estimate. The generalization bound has been used as the objective function for training a deep neural network, whose architecture has been adapted from existing works. Experimental results show state-of-the-art performance of the proposed method (objective function).

**Questions:**

About soundness and contribution (primary):
The bound in Equation (18) is obtained by chaining a sequence of bounds. It seems to me the bounds are loose, for example, Equation (17). So I am curious why it can still outperform the other methods by large gap in the experiments section. And is it because of the use of reweighting schema (GPS) + neural network (a more flexible model), or is it because of using the generalization bounds? It is important to understand which part contributes in the improvement, and to give more explanations on the tightness of the bounds.

About originality:
Are there any literature in causal inference deriving generalization bounds? Not necessarily of a similar form to the one in this work. If so, the authors may want to compare your bound to theirs, and cite them in the paper.




**Limitations:**

The authors do not discuss this.

**Strengths And Weaknesses:**

Originality:
The  proposed bound in this work seems novel to me. One question about related literature on generalization bounds is raised in the next section.

Quality and clarify:
This work is overall of high quality and is very clear in terms of presentation. The only concern I have is about the soundness of the bounds, and is listed in the next section.

Significance:
Judging from the experimental results, the improvements seem significant. However, it is not entirely clear to me why the method can perform so well. The bound seems loose and it is interesting to understand why they can still perform well. A better understanding is needed in order for the method to be more trustworthy.

Some minor remarks:
1. in line 127, \hat u is a typo.
2. in line 198, "the sume of the re-weighted factual loss" -- should it be average?
3. more details are needed for line 201 to line 207;  think the MMD is defined for both continuous and discrete domains, while the authors seems to argue that it cannot be obtained for continuous domains — am I understanding correctly?

---

> ### Author Response · Authors · 2022-08-02
> **Response to Reviewer hMRr**
>
> **Comments 1:**
> 1. $\hat u$ is a typo.
> 2. "the sume of the re-weighted factual loss" -- should it be average? **Yes**
> 3. think the MMD is defined for both continuous and discrete domains, while the authors seems to argue that it cannot be obtained for continuous domains — am I understanding correctly? **Yes**
>
> **Response 1:** Thanks for your careful review. We have thoroughly checked and corrected the grammatical errors and typos we found in the revised manuscript.
>
> For your concern in 3: The understanding is correct. Theoretically, $IPM(t_1,t_2)$ is well defined $\forall t_1,t_2\in(0, 1)$. A certain number of samples are needed to estimate the MMD, as shown in Equation (14). When the treatment is discrete and finite, all the choices of the treatment can be observed in the samples and thus one can estimate the MMD from these samples. However, when the treatment is continuous, the samples that received some treatment $t$ may be unavailable since only a finite number of samples are observed while the choice of $t$ is infinite. We provide a detailed explanation in the revised manuscript.
>
> **Question 1:** The bound in Equation (18) is obtained by chaining a sequence of bounds. It seems to me the bounds are loose, for example, Equation (17). So I am curious why it can still outperform the other methods by large gap in the experiments section. And is it because of the use of reweighting schema (GPS) + neural network (a more flexible model), or is it because of using the generalization bounds? It is important to understand which part contributes in the improvement, and to give more explanations on the tightness of the bounds.
>
> **Answer 1:** Thanks for your helpful comments.
>
> For your concern about Equation (18): The length of the sequence of $IPM_{\Delta max}$ in the algorithm is $L=\lceil\frac1\delta\rceil$. The impact of chaining a sequence of bounds is relatively tiny when $L$ is small ($L=5$ in our experiment).
>
> For your concern about Equation (17): It should be noted that the RHS of Equation (17) represents the bound of the worst case. Without loss of generality, assume $s_1\le s_2\le \cdots\le s_n$, and let $IPM(p_{s_i},p_{s_{i+1}})=\alpha_i|s_{i+1}-s_i|$. It is not difficult to prove that $\alpha$ defined in Assumption 3 satisfies $\alpha=max_{s\in(0,1)}\{\lim_{\delta\to0}\}\frac{IPM(p_s,p_{s+\delta})}{\delta}\ge max_{i\in\{1,2,\cdots,n-1\}}(\alpha_i)$ according to the triangle inequality for the Integral Probability Metric. However, during the proof of Lemmata 2 and 3, all $\alpha_i$ are enlarged to $\alpha$, e.g., the inequality $IPM_\mathcal{G}(p_{s_i}, p_t^w)+IPM_\mathcal{G}(p_{s_i}, p_s)\le IPM_\mathcal{G}(p_{s_i}, p_t^w) + O_p(\frac \alpha {\sqrt[3] n})$ in line 106 of the appendix. This is necessary for the proof, but it also shows that the RHS of Equation (17) represents the bound of the worst case. The bound will be loose when $\alpha$ is large and $\forall i, \alpha_i=\alpha$, i.e., the worst case happens. Intuitively, it is not common for each $\alpha_i$ to be large. For instance, the dose of a particular medicine may depend on the age of the patient, and $\forall i, \alpha_i=\alpha$ means that the age distributions of groups taking similar doses vary considerably, which is unreasonable.
>
> The performance improvement of ADMIT could be attributed to the re-weighting schema and the derived generalization bound that guides a neural network to learn better sampling weights than statistical methods. In our experiments, VCNet quipped with the weights learned by EBCT achieves better performance than the original version of VCNet on two datasets, which demonstrates the effectiveness of re-weighting schema that mitigates the selection bias problem. EBCT estimates sampling weights by solving a globally convex constrained optimization problem but does not provide generalization bounds like our work. Intuitively, the covariates among subpopulations that received different treatments can be balanced by minimizing the $IPM_{\Delta max}$. In other words, the derived bound can guide the learning of superior sampling weights that mitigates the selection bias problem, which is indicated in the experimental results of Table 1.
>
> **Question 2:** Are there any literature in causal inference deriving generalization bounds? Not necessarily of a similar form to the one in this work.
>
> **Answer 2:** Many thanks for your kind reminder. There exists literature that derives generalization bounds for estimating the causal effects of binary treatments, such as [11, 13] cited in our paper. In addition, we also introduce in detail the related studies about the theoretical development in causal inference for binary treatments in a new subsection (A.3) in the revised appendix. We explain the connection between our work and these efforts. On the basis of these studies, to the best of our knowledge, this is the first study that provides a generalization bound for estimating the causal effects of continuous treatments.

---

> > ### Comment · Reviewer_hMRr · 2022-08-06
> > **Author response acknowledgement**
> >
> > Thank you to the authors for the careful response! The answers address my concerns. I read the paper again and still think positively about this paper. I keep my evaluation score unchanged.

---

> > > ### Author Response · Authors · 2022-08-06
> > > **Response to Reviewer hMRr**
> > >
> > > Thank you for your review. We appreciate your acknowledgement of our work!

---

### Meta-Review · Area_Chair_PmDV · 2022-08-26

**Recommendation:** Accept
**Confidence:** Less certain

**Metareview:**

The authors propose theory and an algorithm for estimating average dose-response functions (ADRF) from observational data under assumptions of unconfoundedness and overlap. The approach extends theory and methodology from primarily the work in [13] where neural networks and integral probability metrics are used to learn outcome regressions and re-weighting functions to minimise a bound on the expected loss. The approach was evaluated on semisynthetic datasets and compared favourably to baseline.

Reviewers found the setting novel and interesting but were concerned that the analysis was very close to previous works, requiring only a small modification to allow for continuous (rather than binary) treatments. The empirical evaluation was also rather limited, restricted to comparing mean squared errors on benchmark datasets. One of the reviewers asked why we should expect the method to perform so well when the learning objective represents a fairly loose bound on the expected error. The empirical results offer little to answer this question. The authors rebuttal suggests that this is due to the re-weighting function, but there is no empirical or theoretical evidence that this is the deciding factor. For example, how does the ADMIT model perform without re-weighting? In Figure 3, the authors claim to show that baselines perform worse when selection bias increases, but this trend is noisy at best. If anything, I would argue that it suggests that ADMIT does better no matter the selection bias, which begs the question: where is the advantage coming from?

Overall, reviewers thought the paper appears sound and offered a few clarifying comments and questions which were mostly answered by the authors. The technical novelty is rather low, but appropriately applied. A revised version of the manuscript should address the presentation issues raised by reviewers as well as the attribution question asked above.

**Award:**

No

---

### Decision · Program_Chairs · 2022-09-14

Accept